# Multidrug-Resistant *Helicobacter pylori* Strains: A Five-Year Surveillance Study and Its Genome Characteristics

**DOI:** 10.3390/antibiotics11101391

**Published:** 2022-10-11

**Authors:** Asif Sukri, Alfizah Hanafiah, Hamidah Yusoff, Nur Atiqah Shamsul Nizam, Zarith Nameyrra, Zhiqin Wong, Raja Affendi Raja Ali

**Affiliations:** 1Department of Medical Microbiology & Immunology, Faculty of Medicine, Universiti Kebangsaan Malaysia, Cheras, Kuala Lumpur 56000, Malaysia; 2GUT Research Group, Faculty of Medicine, Universiti Kebangsaan Malaysia, Cheras, Kuala Lumpur 56000, Malaysia; 3Department of Medicine, Faculty of Medicine, Universiti Kebangsaan Malaysia, Cheras, Kuala Lumpur 56000, Malaysia

**Keywords:** *Helicobacter pylori*, multidrug resistance, eradication failure, amoxicillin resistance, genome sequencing

## Abstract

Background: The emergence of multidrug-resistant *Helicobacter pylori* has undermined eradication strategies to prevent the development of gastric cancer. This study was conducted to estimate the prevalence of secondary antibiotic resistance of *H. pylori* in urban multicultural areas in Malaysia. Methods: From January 2017 to December 2021, gastric biopsies from 218 patients with a history of *H. pylori* eradication failure were sent to our laboratory for antibiotic susceptibility testing. A minimal inhibitory concentration was determined for six antibiotics, namely metronidazole, clarithromycin, levofloxacin, amoxicillin, tetracycline, and rifampicin using the E-test method. Two multidrug-resistant *H. pylori* strains identified in this study were subjected to whole genome sequencing. Results: Eradication failure was observed to be significantly higher in the Malaysian Chinese patients than in the Malaysian Indian and Malay patients. *H. pylori* were successfully isolated from 51 patients (23.4%). Overall, the antibiotic resistance rates of *H. pylori* to metronidazole, clarithromycin, levofloxacin, and amoxicillin were 82.4% (42/51), 72.5% (37/51), 52.9% (27/51), and 3.9% (2/51), respectively. Resistance to tetracycline and rifampicin were not observed during the study period. Resistance to more than one antibiotic was observed in 82.4% (42/51) of the isolates, of which 42.2% (19/42) were resistant to three antibiotic classes. Resistance to both clarithromycin and metronidazole were most frequently observed in isolates with dual resistance (56.5%; 13/23). Codon substitutions in penicillin-binding protein 1A (V346L, V374L, G595_V596InsG, Y604H, and N608S) were detected in amoxicillin-resistance *H. pylori* strains. Herein, we report amoxicillin resistance in *H. pylori* isolated from Malaysian patients, and its resistance mechanism, for the first time. Conclusion: Our results show the increase trend in secondary multidrug resistance in *H. pylori* isolates, which warrants continuous surveillance.

## 1. Introduction

The *Helicobacter pylori* infection has been associated with an increased risk of developing gastric cancer and a peptic ulcer [1]. Mortality and incidence of gastric cancer are higher in developing countries, which contributes to high economic and healthcare burdens [2]. The eradication of *H. pylori* using several strategies of an antibiotic regimen is recommended to prevent gastric cancer development [3,4]. Treatment strategies of eradicating *H. pylori* include the initial standard treatment of triple therapy, consisting of a proton pump inhibitor and two antibiotics, namely clarithromycin and metronidazole, in the regions where clarithromycin resistance is low (<15%) [5]. However, the eradication of *H. pylori* using an antibiotic regimen strategy has become complicated since the emergence of multidrug-resistant *H. pylori* strains. The poor adherence of patients to prescribed antibiotics and the overuse of antibiotics, not only in medical healthcare settings but also in agriculture, contribute to the increased rate of antibiotic resistance in bacteria [6]. The efficacy of *H. pylori* eradication using triple therapy is reduced when isolates become resistant to clarithromycin. Therefore, the World Health Organization (WHO) has since listed clarithromycin-resistant *H. pylori* as a high-priority pathogen in the research and discovery of novel antibiotics [7].

In the regions where clarithromycin resistance is high, other treatment options include the following: (1) triple therapy consisting of metronidazole and amoxicillin; (2) non-bismuth concomitant quadruple therapy (PPI, amoxicillin, clarithromycin, and a nitroimidazole); or (3) bismuth-containing quadruple therapy. These treatments are recommended to be administered to the patients instead [7]. The prevalence of *H. pylori* varies according to ethnic group in Malaysia, in which prevalence was high in Malaysian Indians (49.4–52.3%) and Malaysian Chinese people (26.7–57.5%), compared to Malays (11.9–29.2%) [8]. The primary resistance of *H. pylori* is defined as the resistance of the bacteria to antibiotics in patients without a history of an eradication regimen, while secondary resistance is defined as the resistance of the bacteria in patients with a history of eradication treatment. The primary resistance rate to clarithromycin, metronidazole, and levofloxacin in Malaysia ranged from 2.1–12.2%, 36.9–75.5%, and 17.1%, respectively [9]. A larger increase in resistance rate to clarithromycin, metronidazole, and levofloxacin has been observed in secondary isolates than the primary isolates [10]. Meanwhile, no reported primary and secondary resistance isolate has been observed for amoxicillin, tetracycline, and rifampicin in Malaysia [9,10,11,12]. Although the surveillance of antibiotic resistance is important for infection control, there is a lack of data published on the secondary resistance of *H. pylori* in Malaysia. Most of the studies have published data surveillance on the primary resistance of *H. pylori*.

The resistance mechanisms of *H. pylori* to antibiotics include gene mutations, coccoid formation, an overexpression of efflux pumps, and biofilm formation [13]. A recent study in Malaysia has already established the important roles of specific mutations in 23S rRNA, *rdxA*, and *gyrA* genes that confer resistance to clarithromycin, metronidazole, and levofloxacin, respectively [10]. In this study, we report the detection of amoxicillin resistance in *H. pylori* isolated from Malaysian patients for the first time. As no study has been conducted in Malaysia on the mechanisms of amoxicillin resistance in *H. pylori* isolated from Malaysian patients, we adopted the next-generation sequencing platform to investigate the resistance mechanism. The objective of this study was to determine the secondary resistance rates of *H. pylori* in a five-year surveillance period (from 2017–2021) and to elucidate the mechanism of amoxicillin resistance in *H. pylori* isolated from Malaysian patients.

## 2. Results

### 2.1. Demography of Patients

During the study period of January 2017 to December 2021, gastric biopsies were collected from 226 patients. However, eight patients were excluded from this study because of missing demographic data, gastric biopsies that were collected were not properly transported to our laboratory using a *H. pylori* transport medium, and patient with age less than 18 years old during the sample collection. Overall, a total of 218 patients were included in this study. The demographic data of patients included in this study are available in Table 1. The mean age of patients included in this study was 44.5 ± 11.8 years, with ages ranging from 18 to 77 years old. The number of samples collected from patients aged less than 60 years old (87.2%; 190/218) were significantly higher than the number of samples collected from patients aged more than 60 years old (12.8%; 28/218) (*p* < 0.001). No significant difference in gender was observed, in which 51.4% (112/218) of patients were females and 48.6% (106/218) were males. For ethnicity, the majority of samples were collected from Malaysian Chinese people (43.1%; 94/218), followed by Malaysian Indians (22.0%; 48/218), and other ethnic groups (20.2%; 44/218). The lowest frequency of samples was collected from Malaysian Malays (15.1%; 33/218). The samples collected from Malaysian Chinese people were significantly higher than the samples collected from Malaysian Indians and Malays (*p* < 0.001). Overall, *H. pylori* was successfully isolated from 23.4% of patients (51/218).

### 2.2. Antibiotic Resistance Rates

Overall, secondary resistance rates to metronidazole, clarithromycin and levofloxacin during this study period were 82.4% (42/51), 72.5% (37/51), and 52.9% (27/51), respectively (Figure 1). A significant increase in the resistance rate of secondary *H. pylori* isolates to metronidazole was observed from year to year (55.55–100%, *p* = 0.008). An upward resistance trend from year to year, although not significant, was also observed in clarithromycin (55.55–83.33%, *p* = 0.462) and levofloxacin (45.55–58.33, *p* = 0.892).

Throughout the study period, two *H. pylori* isolates (3.9%) were found to be resistant to amoxicillin. One isolate resistant to amoxicillin was isolated from a Malaysian Indian patient in 2017, while the other isolate was isolated from a Malay patient in 2018. From 2019 onwards, no *H. pylori* isolate was observed to be resistant to amoxicillin (Figure 1). No resistance of *H. pylori* to tetracycline and rifampicin was observed during this surveillance study.

### 2.3. Minimal Inhibitory Concentration

The MIC of metronidazole ranged from <0.016 mg/L to >256 mg/L, of which the majority (30/51) of *H. pylori* isolates showed an MIC value >256 mg/L. For clarithromycin, the MIC value ranged from <0.016 mg/L to >256 mg/L, of which the majority (51%; 26/51) of isolates demonstrated an MIC value ≥6 mg/L. Meanwhile, the majority of isolates (70.6%; 36/51) that were susceptible to amoxicillin exhibited an MIC value ≤0.016 mg/L, while two resistant isolates exhibited an MIC value of 48 mg/L and >256 mg/L, respectively. While MIC for levofloxacin ranged from <0.002 mg/L to >32 mg/L, the majority of resistant isolates (66.67%; 18/27) had an MIC value >32 mg/L (Appendix A).

### 2.4. Secondary Multidrug Resistance

During the 5-year surveillance period, 80.4% (41/51) of the secondary isolates successfully cultured in this study were multidrug-resistant isolates. The majority of isolates (46.34%; 19/41) were triple resistant, of which 94.73% (18/19) of the isolates were resistant to metronidazole, clarithromycin, and levofloxacin, while one isolate (5.27%) was resistant to metronidazole, clarithromycin, and amoxicillin. Dual resistance to metronidazole and clarithromycin was observed in 36.6% (15/41) of the isolates, while resistance to metronidazole and clarithromycin was recorded in 14.6% (6/41) of the isolates. One isolate was found to be resistant to both metronidazole and amoxicillin (Table 2).

### 2.5. Genome Sequencing of Multidrug-Resistant H. pylori

The genomes of two multidrug-resistant *H. pylori* (designated as HP18045) isolated in our study were sequenced to elucidate a resistance mechanism. The quality of *H. pylori* genome sequencing is available in Appendix A. HP18045 was assembled into 34 contigs (1.63 Mb, N50 133 kb, 38.81% GC), while HP21041 was assembled into 41 contigs (1.61 Mb, N50 93 kb, 39.21% GC). Genome coverage of both strains was >500×. The scanning of both genome assemblies for 628 highly conserved bacterial protein-coding genes showed that both genome assemblies had a genome completeness of >99% with a zero-duplication rate, indicating high-quality bacterial genome assemblies. HP18045 and HP21041 contained 1535 and 1488 coding sequences, respectively. Both strains had 2 rRNAs and 36 tRNAs. We found 660 genes that encode hypothetical proteins in HP18045 and 628 genes that encode hypothetical proteins in HP21041.

### 2.6. Virulence Factors Detected in Multidrug-Resistant H. pylori

The genomes of both samples were screened for the detection of important virulence factors in *H. pylori*. *cagA* was detected in HP18045, while *cagA* was not detected in HP21041. The characterization of Glu-Pro-Ile-Tyr-Ala (EPIYA) *cagA* repeats in HP18045 showed that it belonged to the EPIYA-AB type. *cagS*, *cagT*, and genes that make up the type IV secretion system (T4SS), namely *virB4*, *virB11*, and *ptlH*, were detected in HP18045. Meanwhile, genes that make up the T4SS, namely *virB4*, *virB11*, *ptlC*, and *ptlH*, were detected in HP21041. Other *cag*PAI virulence genes were not detected in the HP21041 strain. Both samples harbored insertion sequences (IS) of IS200/605 family transposase elements, in which HP18045 harbored the IS605 element while HP21041 harbored IS608. HP18045 harbored the *vacA* s1m2 allele, while HP21041 harbored the *vacA* s2m2 allele. Outer inflammatory protein A (OipA) was not detected in either strain.

### 2.7. Antibiotic-Resistant Genes in Amoxicillin-Resistant H. pylori

We adopted the RGI online bioinformatic tool available in the CARD database to predict resistomes from genome sequences of the multidrug-resistant *H. pylori* assessed in this study. A list of antibiotic resistance genes, their mutations, mechanisms of antibiotic resistance, and the percentage of identities matching the regions detected in HP18045 and HP21041 is available in Appendix A. HP18045 was predicted to harbor seven antibiotic-resistant genes, while HP21041 was predicted to harbor six. The mutations of antibiotic-resistant genes predicted to occur in both strains included the mutations of *pbp1* and *pbp2* (codons S494H and E572G), conferring resistance to amoxicillin, *rpoB* codon mutation (K2068R), conferring resistance to rifampicin, and 23S *rRNA* mutations (C1707T and A2144G), conferring resistance to clarithromycin. The mutations of *frxA* (Y62D) and *rdxA* (R90K, A118S, C49T and D59N), conferring resistance to metronidazole, were detected in HP18045, while in HP21041, those mutations were not detected. Meanwhile, the mutations of *gyrA* (codon T87I and N87I) that conferred resistance to fluoroquinolone were observed in HP21041, but not in HP18045. *vanT* and *vanTr* mutations, conferring resistance to glycopeptide, were also detected in HP18045 and HP21041, respectively, but the percentage of the protein sequence identity in comparison to the sequences available in the database was <36%.

To elucidate the mechanism of amoxicillin resistance, we screened variants of the PBP1A protein sequence of the amoxicillin-resistant strain in our study. A list of codon variants observed in amoxicillin-resistant and amoxicillin-susceptible strains is shown in Table 3. Overall, 12 variants were detected in the PBP1A protein sequence of amoxicillin-resistant strains, of which seven variants (V16I, F125L, I148L, G242S, N322D, D508E, and D535’) were detected in both amoxicillin-resistant (HP18045) and amoxicillin-susceptible (HP21041) variants. Three variants were identified at the transglycosylase domain, while seven variants were identified at the transpeptidase domain. Five variants, namely V346L, V374L, G595_V596InsG, Y604H, and N608S, were exclusively identified in the amoxicillin-resistant strain, but not in the amoxicillin-susceptible strain.

### 2.8. Genotypic and Phenotypic Correlation of the MDR H. pylori Strains

Next, we examined the phenotypic and genotypic correlation of the MDR *H. pylori* subjected to whole-genome sequencing in our study. HP21041 was resistant to three drugs, namely metronidazole, clarithromycin, and levofloxacin, while H18045 was resistant to three drugs, namely metronidazole, clarithromycin and amoxicillin. Based on the genome analysis, the genotypic and phenotypic correlation in HP18045 was consistent for amoxicillin (mutation of *pbp1A*), metronidazole (mutations of *frxA* and *rdxA*), and clarithromycin (mutation of 23S rRNA) resistance. The mutation of *rpoB* (rifampicin) was also observed at the genotypic level, although the strain was susceptible to the antibiotic at the phenotype level (Appendix A). Meanwhile, the genotypic result of HP21045 was consistent with the phenotypic result of levofloxacin and clarithromycin resistance, but not metronidazole, amoxicillin, and rifampicin. The sample harbored mutations of resistance gene *gyrA* and 23S rRNA, which is important in levofloxacin and clarithromycin resistance, respectively. Although HP21045 was predicted to harbor the resistance gene mutation of *rpoB* and *pbp1A*, conferring resistance to rifampicin and amoxicillin, the resistance was not observed at the phenotypic level. Additionally, the mutation of the resistance gene that conferred resistance to metronidazole was not detected in HP21045, although the strain was resistant to metronidazole at the phenotypic level (Appendix A).

## 3. Discussion

Malaysia is a multicultural country with Malays, Malaysian Chinese people, and Malaysian Indians comprising the three largest ethnicities in the country. Although gastric cancer is the twelfth most common cancer in Malaysia, the difference of gastric cancer incidence among the three major ethnicities in this country has been observed to be significantly different [14]. The incidence of gastric cancer is significantly higher in Malaysian Chinese people compared to Malaysian Indians and Malays [15]. Furthermore, the gastric adenocarcinoma cardia subtype was more frequently detected in Malaysian Chinese patients than in Malay patients [16], suggesting a different etiology of gastric carcinogenesis between Chinese and Malay patients. In this study, we found that eradication failure was observed to be significantly higher in Malaysian Chinese patients than Malaysian Indian and Malay patients. Consistent with observation from previous studies that Malaysian Chinese patients have a significantly higher risk of developing gastric cancer [15], the difficulty of eradicating *H. pylori* among Malaysian Chinese patients compared to other ethnicities may contribute to this phenomenon and should be studied further. *H. pylori* isolates from patients with different ethnicities in Malaysia harbored different virulence factors that rendered them resistant to antibiotics [17]. For instance, the isolates from Malaysian Chinese patients harbored East Asian *cagA*, while the isolates from the Malays and the Malaysian Indians harbored Western *cagA*. East Asian *cagA* has been observed to be more pathogenic than Western *cagA* [18]. We also observed eradication failure tends to occur more frequently in patients aged less than 60 years old than that in patients aged more than 60 years old. A recent study on a large population also observed that the eradication failure of *H. pylori* occurs more in younger patients than in older patients [19]. This trend should be studied further as a failure of eradication in young people will complicate the early eradication therapy of *H. pylori* to prevent gastric cancer development. Overall, we successfully isolated *H. pylori* from 23.4% of the gastric biopsies sent to our laboratory. *H. pylori* is difficult to culture due to the nature of bacteria in that it is slow-growing and fastidious. However, culture is still considered to be the gold standard in the diagnosis of *H. pylori* and antibiotic susceptibility purposes. One limitation of our study is that we did not perform a molecular analysis of *H. pylori* for specific mutation detection directly from the gastric biopsies for clarithromycin and levofloxacin resistance. Future work should perform molecular analysis from gastric biopsies with unsuccessful cultures of *H. pylori*.

The secondary resistance rates to metronidazole, clarithromycin, and levofloxacin were high during the period of surveillance, with a significant upward trend observed for metronidazole from 2017 to 2021. The secondary resistance rates to metronidazole, clarithromycin, and levofloxacin were observed to be high worldwide [20]. In this study, we observed the trend in resistance rates to metronidazole, clarithromycin, and levofloxacin, which increased from the previous study period [10]. The distribution of MIC values revealed that the majority of isolates demonstrated high MIC values against levofloxacin, clarithromycin, and metronidazole. The increase in MIC values for levofloxacin is a major concern as it is used as an option for second-line treatment and the bacteria can rapidly acquire resistance to quinolone [21].

For the first time, we detected amoxicillin resistance in a secondary isolate from Malaysian patients in 2017 and 2018. No amoxicillin resistance in both the primary and secondary isolates has been observed in previous studies in Malaysia [9]. Worldwide, secondary resistance to amoxicillin is still low [20]. Consistent with previous findings [10,11,12], no secondary isolates were resistant to tetracycline and rifampicin throughout our surveillance study.

High triple-resistance rates to clarithromycin, levofloxacin, and metronidazole were observed in our study. Furthermore, a high dual-resistance rate to clarithromycin and metronidazole was also recorded, indicating a major public health concern as clarithromycin and metronidazole are used for sequential, hybrid, and concomitant therapies [5]. In European countries, the secondary resistance rate to clarithromycin and metronidazole was found to be 18%, while in our study, it was found to be 36.6% [19]. Culture-guided treatments and genetics analyses of *H. pylori* should be conducted in the future to determine the antibiotic susceptibility of *H. pylori* before the administration of treatment.

We attempted to elucidate the antibiotic resistance mechanisms of multidrug-resistant *H. pylori* in our study using genomic analysis. Contrary to the previous study [17], harboring virulence factors, namely *cagA*, *vacA*, and *cag*PAI genes, did not play an essential role in conferring multidrug resistance to *H. pylori*. However, this finding should be interpreted with caution as we only examined the genomes of two resistant strains in our study. The correlation between phenotypic and genotypic antibiotic resistance was found in amoxicillin, clarithromycin, levofloxacin, and metronidazole resistance. Interestingly, a discrepancy in the genotypic and phenotypic correlation was discovered in rifampicin resistance. The *rpoB* gene mutation that is associated with rifampicin [13] was detected in both *H. pylori* strains examined in this study, although both strains were susceptible to rifampicin at the phenotypic level. This finding suggests that the rifampicin resistance mechanism in *H. pylori* can occur through other mechanisms, namely the post-translational modification of protein and the presence of other antibiotic-resistant genes. An alteration in the structure of the PBP1A protein leads to a reduced binding of amoxicillin to the target protein and subsequently protects the bacteria from cell-wall damage caused by amoxicillin [13]. Codon substitutions of the PBP1A protein were identified in amoxicillin-resistant *H. pylori* in this study. Two out of five codon variants detected in amoxicillin resistance in this study, namely V374L and G595_V596InsG, were previously shown to confer resistance to amoxicillin in *H. pylori* isolated from Cambodia [22] and Vietnam [23]. The rest of the codon variants, namely V346L, Y604H, and N608S, detected in amoxicillin-resistant *H. pylori* in this study were not previously reported in the literature. Herein, we reported, for the first-time, the amoxicillin resistance mechanism of *H. pylori* isolated from Malaysian patients through codon substitutions in the PBP1A protein.

## 4. Materials and Methods

### 4.1. Sample Collection

Gastric biopsies were sent for *H. pylori* antibiotic-susceptibility testing as a routine diagnostic service; therefore, ethics approval is not applicable for this study. From January 2017 to December 2021, gastric biopsies from patients who failed the *H. pylori* eradication regimen from healthcare centers in the urban area of Klang Valley, Malaysia were sent to the Bacteriology Laboratory, Department of Medical Microbiology and Immunology, Faculty of Medicine, Universiti Kebangsaan Malaysia for antibiotic-susceptibility testing. The biopsies were taken from antrum and/or corpus, and immediately placed into the *H. pylori* transport medium with 15% glycerol and transported to the lab within 6 h at a cold temperature.

### 4.2. Culture of H. pylori

Gastric biopsies were streaked onto Columbia blood agar (Oxoid, Basingstoke, UK) plates that consisted of 7% defibrinated sheep blood to isolate *H. pylori* colonies. The plates were incubated at 37 °C for 5–10 days in microaerophilic conditions. A positive *H. pylori* culture was confirmed using biochemical tests (oxidase and urease tests) and a morphological examination (Gram staining). Colonies of *H. pylori* were subcultured for antibiotic-susceptibility testing and stock cultures were prepared and stored at −80 °C.

### 4.3. Antibiotic Susceptibility Testing

We determined the antibiotic susceptibility of *H. pylori* to six classes of antibiotics, namely metronidazole, clarithromycin, amoxicillin, levofloxacin, tetracycline, and rifampicin. The antibiotic susceptibility testing was performed using E-test strips (bioMérieux, Marcy-l’ Étoile, France) as described previously [10]. The antibiotic resistance breakpoints were >8 mg/L for metronidazole [24], >4 mg/L for rifampicin [25], and >1 mg/L for amoxicillin [26], levofloxacin [27], and tetracycline [28]. According to the Clinical Laboratory Standard Institute (CLSI), the resistance cut-off point for clarithromycin was ≥1 mg/L; meanwhile, to be defined as clarithromycin-susceptible and intermediate, the cut-off points were ≤0.25 mg/L and 0.5 mg/L, respectively [29]. Multidrug resistance is defined as resistance to two or more classes of antimicrobial agents.

### 4.4. Genome Sequencing of Multidrug-Resistant H. pylori

The genomes of two multidrug-resistant *H. pylori* isolates (designated as HP18045 and HP21041) were sequenced. HP18045 was resistant to amoxicillin, metronidazole, and clarithormycin, while HP21041 was susceptible to amoxicillin but resistance to metronidazole, clarithromycin, and levoflaxacin. The genomic DNA of *H. pylori* was extracted using the NucleoSpin Microbial DNA Mini kit as described by the manufacturer (Macherey-Nagel, Düren, Germany). The integrity of the extracted DNA was analyzed on 0.5% agarose gel. Approximately 100 ng of DNA, as measured by Qubit, was fragmented to 350 bp using a Bioruptor followed by NEB Ultra II library preparation according to the manufacturer’s instructions (NEB, Ipswich, MA, USA). Sequencing was performed on a NovaSEQ6000 (Illumina, San Diego, CA, USA), generating approximately 1 Gb of paired-end data (2 × 150 bp) for each sample. Raw paired-end reads were quality-, poly-G-, and adapter-trimmed using fastp v0.21.0 [30], retaining filtered paired-end reads that were longer than 125 bp. De novo assembly was performed using unicycler v. 0.4.8 [31], and contigs longer than 500 bp were reported and used for subsequent analysis. The assembly statistics calculation and the genome completeness calculation used QUAST v5 and BUSCO v. 5.3.0 (Campylobacterales odb10), respectively [32,33]. Gene annotation was conducted using Prokka v. 1.14.6 [34]. The genomic data of this study were deposited into the GenBank database, National Centre of Biotechnology Institute (NCBI) [35] under accession number: JANTPN000000000 (*H. pylori* HP18045) and JANTPO000000000 (*H. pylori* HP21041).

### 4.5. Analysis of Antibiotic Resistance Genes and Virulence Factors

We examined the presence of antibiotic-resistant genes from *H. pylori* genomic sequence data using the Comprehensive Antibiotic Resistance Database (CARD) [36]. The resistomes of multidrug-resistant *H. pylori* was determined using the Resistance Gene Identifier (RGI) tool available in the CARD database. Furthermore, the presence of *H. pylori* virulence factors, namely cytotoxin-associated gene A (*cagA*), cytotoxin-associated gene pathogenicity island (*cag*PAI) genes, and outer membrane proteins in the genomes, was also assessed. The analysis of the VacA protein region signal (s) and middle (m) was performed by comparing protein sequences to the VacA sequences of the Tx30a strain (s2m2; GenBank no. U29401) and the NCTC 11638 strain (s1m1; GenBank no. GenBank: U07145). As amoxicillin resistance is associated with mutations at penicillin-binding protein 1A (*pbp1A*) [26], we extracted the DNA and protein sequences of the *pbp1A* gene of amoxicillin-resistant *H. pylori* in our study and compared the sequence to the *pbp1A* sequence of the reference strain (*H. pylori* 26695, GenBank accession no: AE000511) and the HP21041 strain using the Clustal Omega platform [37]. To detect the presence of a possible novel variant in the PBP1A protein of amoxicillin resistance in *H. pylori*, we examined the variants of the PBP1A protein reported so far in the CARD database [36], as well as in scientific publications, and compared those reported variants with the protein variants detected in our study.

### 4.6. Statistical Analysis

Categorical variables in demographic data, including age, ethnicity and gender, were analyzed using Pearson’s chi-square test. The trend in antibiotic resistance from year to year (2017 to 2021) was analyzed using Pearson’s chi-square test, while independent factors related to the development of antibiotic resistance was analyzed using linear regression analysis. All analyses were conducted using Statistical Package for the Social Science (SPSS) software version 22.

## 5. Conclusions

The high eradication failure levels observed among Malaysian Chinese people compared to other major ethnicities likely explains the high prevalence of gastric cancer among Malaysian Chinese people compared to other ethnicities in Malaysia. We observed high secondary resistance rates to metronidazole, clarithromycin, and levofloxacin with an upward trend, indicating the urgency for proper infection control. For the first time, we detected the amoxicillin resistance of *H. pylori* isolated from Malaysian patients. Furthermore, we also interrogated the genomes of amoxicillin-resistant strains and revealed codon substitutions in the PBP1A protein as a contributor to amoxicillin resistance in our study. We will continue to monitor antibiotic resistance patterns of *H. pylori* in our center and apply genomic analysis as part of our surveillance efforts. The emergence of multidrug-resistant *H. pylori* in Malaysia among patients with eradication failure warrants an intervention strategy from clinicians and policymakers.

## Figures and Tables

**Figure 1 antibiotics-11-01391-f001:**
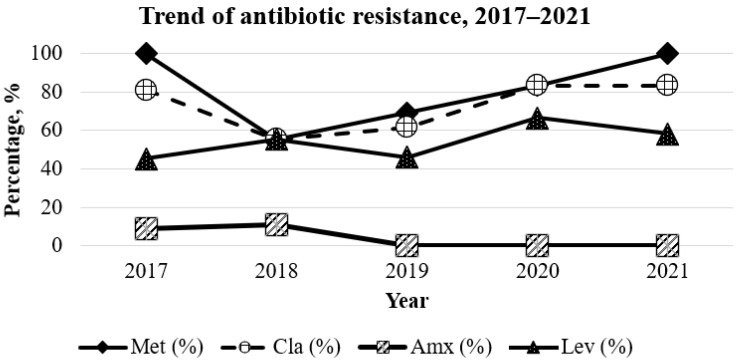
Trend in secondary antibiotic resistance from 2017–2021.

**Table 1 antibiotics-11-01391-t001:** Demography of patients.

	Overall (*n* = 218)	*p*-Value
Age:		
Range	18–77 years	
Mean ± SD	44.5 ± 11.8 years	
Age group:		<0.001 *
<60 years	190 (87.2)	
>60 years	28 (12.8)	
Gender, *n* (%):		
Male	106 (48.6)	
Female	112 (51.4)	
Ethnicity:		<0.001 *
Malay	33 (15.1)	
Chinese	94 (43.1)	
Indian	48 (22.0)	
Other	43 (20.2)	
Positive culture, *n* (%)	51 (23.4)	

* Pearson’s chi-square test.

**Table 2 antibiotics-11-01391-t002:** Multidrug resistance of the secondary *H. pylori* isolates from 2017–2021 (*n* = 41).

Multidrug Resistance	2017, (*n* = 11) -%-	2018, (*n* = 9) -%-	2019, (*n* = 13) -%-	2020, (*n* = 6) -%-	2021, (*n* = 12) -%-	Total, (*n* = 41) %
MTZ, CLA	45.5	11.1	7.7	16.7	58.33	36.6
MTZ, LEV	9.1	44.4	0	0	8.3	14.6
MTX, AMO	9.1	0	0	0	0	2.4
MTZ, CLA, LEV	36.4	11.1	46.2	66.7	25	43.9
MTZ, CLA, AMO	0	11.1	0	0	0	11.1

MTZ: metronidazole; CLA: clarithromycin; LEV: levofloxacin; AMO: amoxicillin.

**Table 3 antibiotics-11-01391-t003:** List of codon variants detected in amoxicillin-resistance and amoxicillin-susceptible *H. pylori* strains.

Variant	Detection of PBP1A Variants	
	HP18045	HP21041
V16I	Yes	Yes
F125L	Yes	Yes
I148L	Yes	Yes
G242S	Yes	Yes
N322D	Yes	Yes
V346L	Yes	No
V374L	Yes	No
D508E	Yes	Yes
D535N	Yes	Yes
G595_V596InsG	Yes	No
Y604H	Yes	No
N608S	Yes	No

HP18045 was resistant to amoxicillin, while HP21041 was susceptible. “Yes” denotes variant was detected in *H. pylori* strain, while “No” denotes variant was not detected in *H. pylori* strain examined. Protein sequence comparison was made with reference strain *H. pylori* 26695. PBP1A: penicillin-binding protein 1A.

## Data Availability

All data from this study are available in the article and the Appendix A.

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
