# Peer review of "Multidrug-Resistant Helicobacter pylori Strains: A Five-Year Surveillance Study and Its Genome Characteristics"

_antibiotics, 2022, doi:10.3390/antibiotics11101391_

Round 1

Reviewer 1 Report

1. Trivial considedartions and well-known data on Helicobacter pylori could be deleted from the introduction section

2. The background prevalence of H. pyloria in Malaysia must be given in values and detailed according to ethnic/religion groups.

3. The correlation between presence/absence of virualence facvtoers and AB resistance should be given (cagA, etc)

4. If a bacterium has genetic jutation of rifabutin gene, but is susceptibile in culture to this drug, is it considered senssitive or resistant to this drug?

5. The ciscussion section seems to be redundant, resembling rather a review on AB resistance. 

Author Response

Reviewer 1:

  1. Trivial considerations and well-known data on Helicobacter pylori could be deleted from the introduction section

Thank you for this comment. We agree with this comment and deleted line 41-43, 59-64 because it is trivial and well-known fact about antibiotics.

  1. The background prevalence of H. pylori in Malaysia must be given in values and detailed according to ethnic/religion groups.

Thank you for this comment. We included the background prevalence of H. pylori in Malaysia according to ethnic groups in Introduction section (line 68-70) as suggested.

  1. The correlation between presence/absence of virualence factors and AB resistance should be given (cagA, etc)

Thank you for this suggestion. Due to resource constraint, we only managed to sequence genomes of two H. pylori strains. Thus, correlation between presence or absence of virulence factors could not be statistically calculated. However, we found that cagA EPIYA motif, the presence of all cagPAI genes and vacA polymorphism did not influence antibiotic resistance in these two strains examined.

  1. If a bacterium has genetic mutation of rifabutin gene, but is susceptible in culture to this drug, is it considered sensitive or resistant to this drug?

No, if the bacterium harboured resistance gene to certain drug but susceptible to that drug in culture, it would be considered as susceptible. We already explained the criteria that we undertook for H. pylori strains to be considered as resistant/susceptible in Materials and Methods section (section 4.3).

  1. The discussion section seems to be redundant, resembling rather a review on AB resistance. 

Thank you for this comment. We deleted unnecessary sentences that we thought we already mentioned in the Introduction section to make our Discussion section more concise.

Reviewer 2 Report

This study was conducted to estimate the prevalence of secondary antibiotic resistance of H. pylori at urban multicultural area in Malaysia.

1. During the study period from January 2017 to December 2021, H. pylori were successfully isolated from 51 patients (23%).

The number of patients enrolled in this study was very small and the selection bias could not be avoided.

2. As shown in Table 1, the range of age was from 10 months to 77 years old.

It meant that the age of one patient was only 10 months and he/she had the history of H. pylori eradication failure. This is hard to be imagined. Even if this is true, the data can easily be questioned.

3. Figure 1: Trend of secondary antibiotic resistance from 2017-2021

Fifty-one patients were included and only two H. pylori isolates (3.9%) were found to be resistant to amoxicillin. Therefore, it is obvious that the result for the trend may be inaccurate.

4. Table 2: Multidrug resistance of the secondary H. pylori isolates from 2017-2021

Only 41 patients were included and the trend was inaccurate.

5. The genomes of two multidrug-resistant H. pylori (designated as HP18045 and HP21041) isolated in our study were sequenced to elucidate resistance mechanism.

It was not enough that only two strains were used.

6. Codon substitutions in penicillin-binding protein 1A (V346L, V374L, G595_V596InsG, Y604H, and N608S) were detected in amoxicillin-resistance H. pylori strain. Herein we report for the first-time amoxicillin resistance in H. pylori isolated from Malaysian patients and its resistance mechanism.

Mutations of PBP1 associated with amoxicillin resistance have been reported by many previous studies.

For example:

Paul R, Postius S, Melchers K, Schäfer KP. Mutations of the Helicobacter pylori genes rdxA and pbp1 cause resistance against metronidazole and amoxicillin. Antimicrob Agents Chemother. 2001 Mar;45(3):962-5. doi: 10.1128/AAC.45.3.962-965.2001. PMID: 11181392; PMCID: PMC90405.

Okamoto T, Yoshiyama H, Nakazawa T, Park ID, Chang MW, Yanai H, Okita K, Shirai M. A change in PBP1 is involved in amoxicillin resistance of clinical isolates of Helicobacter pylori. J Antimicrob Chemother. 2002 Dec;50(6):849-56. doi: 10.1093/jac/dkf140. PMID: 12461003.

Author Response

Reviewer 2:

This study was conducted to estimate the prevalence of secondary antibiotic resistance of H. pylori at urban multicultural area in Malaysia.

  1. During the study period from January 2017 to December 2021, H. pylori were successfully isolated from 51 patients (23%).

The number of patients enrolled in this study was very small and the selection bias could not be avoided.

Thank you for this comment. We enrolled 218 patients with history of eradication failure into our study. Proportion of eradication failure in Malaysia is 16.3% based on our previous finding and according to our sample size calculation, we need to collect gastric biopsies from at least 214 patients (confidence level=95% and margin of error=5%) (https://www.calculator.net/sample-size-calculator.html). Culture of H. pylori is challenging as it is a fastidious microorganism and consistent with our previous study, success rate was ~20.4% from patients with dyspepsia symptoms (Hanafiah A et al. Molecular characterization and prevalence of antibiotic resistance in Helicobacter pylori isolates in Kuala Lumpur, Malaysia. Infect Drug Resist. 2019; 12:3051-3061. doi: 10.2147/IDR.S219069). Furthermore, H. pylori culture success rate tends to drop for patients with history of eradication failure. Nevertheless, culture-guided therapy is still a preferable method for successful H. pylori.

  1. As shown in Table 1, the range of age was from 10 months to 77 years old. It meant that the age of one patient was only 10 months and he/she had the history of H. pylori eradication failure. This is hard to be imagined. Even if this is true, the data can easily be questioned.

Thank you for this comment. We agree with this comment and decided to exclude the patients who were less than 18 years old during sample collection from our study. Thus, total of samples included in our statistical analysis became 218 patients and there is a change in analysis of demographic data such as ethnicity and age (Table 1). However, no change in statistical analysis of antibiotic resistance rate because H. pylori was not successfully isolated from the sample collected from all patients who were less than 18 years old.

  1. Figure 1: Trend of secondary antibiotic resistance from 2017-2021. Fifty-one patients were included and only two H. pylori isolates (3.9%) were found to be resistant to amoxicillin. Therefore, it is obvious that the result for the trend may be inaccurate.

Thank you for this comment. This study was conducted as surveillance program to determine antibiotic resistance rate in patients with H. pylori eradication failure. Because H. pylori isolation from gastric biopsies can be challenging due to the nature of the bacterium itself that is slow growing and fastidious, we managed to successfully isolate 51 (23.4%) strains of H. pylori from gastric biopsies isolated from 218 patients. This is because the success rate of H. pylori culture decreases for gastric biopsies obtained from patients with history of eradication failure. However, we successfully detected the presence of amoxicillin resistance in H. pylori isolated from our study, which is the first report in Malaysia. This new report indicates that while the success rate of H. pylori culture is low, we still managed to report the new finding of amoxicillin resistance H. pylori isolated from Malaysian patient throughout our 5-year surveillance study.

  1. Table 2: Multidrug resistance of the secondary H. pylori isolates from 2017-2021. Only 41 patients were included and the trend was inaccurate.

Thank you for this comment. As mentioned in responses to Comments no. 2 and 3, this study collected gastric biopsies from patients with history of eradication failure for 5-year duration. As prevalence of patients with eradication failure in Malaysia was 16.3% based on our previous study, we calculated the sample size and came out with at least 214 patients need to be recruited in our study. Culture of H. pylori is challenging and difficult especially for gastric biopsies collected from patients with history of eradication failure. Thus, we only managed to successfully isolate 51 strains out of 218 gastric biopsies collected. However, our data are still meaningful because studies on antibiotic resistance in secondary isolates from Malaysian patients are scarce.

  1. The genomes of two multidrug-resistant H. pylori (designated as HP18045 and HP21041) isolated in our study were sequenced to elucidate resistance mechanism. It was not enough that only two strains were used.

Thank you for this comment. Due to resource constraint, we only managed to sequence two H. pylori strains. However, the data that we obtained are useful because this is the first time that genome of amoxicillin resistance H. pylori isolated from Malaysian patient was sequenced. The results shed the light on the mechanism of amoxicillin resistance in H. pylori isolated from Malaysian patient. Furthermore, there has been no report on genome sequence of amoxicillin resistance H. pylori isolated from Malaysian patient and the data that we obtained from this study has been deposited to public database that can be accessed for further analysis and collaboration in the future.

  1. Codon substitutions in penicillin-binding protein 1A (V346L, V374L, G595_V596InsG, Y604H, and N608S) were detected in amoxicillin-resistance H. pylori strain. Herein we report for the first-time amoxicillin resistance in H. pylori isolated from Malaysian patients and its resistance mechanism. Mutations of PBP1 associated with amoxicillin resistance have been reported by many previous studies.

For example:

Paul R, Postius S, Melchers K, Schäfer KP. Mutations of the Helicobacter pylori genes rdxA and pbp1 cause resistance against metronidazole and amoxicillin. Antimicrob Agents Chemother. 2001 Mar;45(3):962-5. doi: 10.1128/AAC.45.3.962-965.2001. PMID: 11181392; PMCID: PMC90405.

Okamoto T, Yoshiyama H, Nakazawa T, Park ID, Chang MW, Yanai H, Okita K, Shirai M. A change in PBP1 is involved in amoxicillin resistance of clinical isolates of Helicobacter pylori. J Antimicrob Chemother. 2002 Dec;50(6):849-56. doi: 10.1093/jac/dkf140. PMID: 12461003.

Thank you for this comment. We apologize for confusion caused. What we were trying to convey in the paragraph is that no previous studies conducted in Malaysia have found amoxicillin resistance in H. pylori isolated from Malaysian patients. To the best of our knowledge, our study is the first one to detect amoxicillin resistance in H. pylori isolated from Malaysian patient. While we are aware that many previous studies have discovered mutations of PBP1 protein as mechanism that confers resistance to amoxicillin, no study has been conducted to study mutation of PBP1 protein in amoxicillin-resistant H. pylori from Malaysian patient as no amoxicillin-resistant H. pylori has been isolated from Malaysian patient. In this study, we found mutations of PBP1 in amoxicillin-resistant H. pylori that confer resistance to amoxicillin from the strains isolated from Malaysian patients.

Reviewer 3 Report

Dear Author,

really very interesting this article, and very a fantastic result on the resistance of H.P

Many thanks

Author Response

Reviewer 3:

Dear Author,

really very interesting this article, and very a fantastic result on the resistance of H.P Many thanks

Thank you so much.

Round 2

Reviewer 2 Report

I recommend that it can be accepted for publication in present form.